# Self-assessment of unilateral and bimodal cochlear implant experiences in daily life

Elke M. J. Devocht[1]*, A. Miranda L. Janssen[1,2], Josef Chalupper[3], Robert J. Stokroos[1¤], Herman Kingma[4], Erwin L. J. George[1]

1 Department of ENT/Audiology, School for Mental Health and Neuroscience (MHENS), Maastricht University Medical Center, Maastricht, The Netherlands, 2 Department of Methodology and Statistics, School for Public Health and Primary Care (CAPHRI), Maastricht University, Maastricht, The Netherlands, 3 Advanced Bionics European Research Centre, Hannover, Germany, 4 Department of ENT, School for Oncology and Develop Biology (GROW), Maastricht University Medical Center, Maastricht, The Netherlands

¤ Current address: Department of ENT/Brain Center Rudolf Magnus, University Medical Center Utrecht, Utrecht, The Netherlands

* elke.devocht@mumc.nl

## Abstract

### Objective

The subjective experiences were assessed of cochlear implant (CI) users either wearing or not wearing a hearing aid (HA) at the contralateral ear.

### Design

Unilateral CI-recipients were asked to fill out a set of daily-life questionnaires on bimodal HA use, hearing disability, hearing handicap and general quality of life.

### Study sample

Twenty-six CI-recipients who regularly use a contralateral HA (bimodal group) and twenty-two CI-recipients who do not use a HA in the contralateral ear (unilateral group).

### Results

Comparisons between both groups (bimodal versus unilateral) showed no difference in self-rated disability, hearing handicap or general quality of life. However within the group of bimodal listeners, participants did report a benefit of bimodal hearing ability in various daily life listening situations.

### Conclusions

Bimodal benefit in daily life can consistently be experienced and reported within the group of bimodal users.

**Data Availability Statement:** All relevant data are within the manuscript and its Supporting Information files.

**Funding:** A research grant from Advanced Bionics Inc. to MUMC+ financially supported the work of the first author (E.D.) in this investigator-initiated study. The second author (M.J.) reviewed the manuscript and provided statistical support, made possible by a grant from the Dutch Heinsius-Houbolt Foundation. The third author (J.C.) had a role in designing the study. He holds a scientific post in the Advanced Bionics European Research Center. For the remaining authors no conflicts were declared. The study was designed in cooperation between MUMC+ and Advanced Bionics. Data collection, analysis and the decision to publish were all solely accounted for by MUMC+. The work presented in this manuscript is the intellectual property of MUMC+.

**Competing interests:** A research grant from Advanced Bionics Inc. to MUMC+ financially supported the work of the first author (E.D.) in this investigator-initiated study. The second author (M. J.) reviewed the manuscript and provided statistical support, made possible by a grant from the Dutch Heinsius-Houbolt Foundation. The third author (J. C.) had a role in designing the study. He holds a scientific post in the Advanced Bionics European Research Center. For the remaining authors no conflicts were declared. The study was designed in cooperation between MUMC+ and Advanced Bionics. Data collection, analysis and the decision to publish were all solely accounted for by MUMC +. The work presented in this manuscript is the intellectual property of MUMC+.

## Introduction

Given the beneficial results of cochlear implantation (CI), candidacy criteria for receiving a CI have broadened from profound hearing loss to also include moderately-severe hearing loss [1–3]. As a result a growing number of CI recipients have aidable residual hearing in the contralateral ear [4]. In many countries around the world, current reimbursement regulations for adults support solely unilateral cochlear implantation. Fitting a contralateral hearing aid in the non-implanted ear when aidable residual hearing is present is generally recommended and well established as standard clinical practice [5–7]. Previous research has shown that an increasing number (50 to 60%) of CI recipients who receive a CI in one ear indeed prefer to retain their acoustic HA in the non-implanted ear [8–10].

Combining electrical stimulation by a CI in one ear with acoustic amplification by a conventional HA in the other ear, is known by the label of bimodal hearing. The benefits of bimodal hearing are attributed to the combined effects of the use of two ears (bilateral input), the opportunity to centrally combine the input in both ears (binaural cues), and the access to complementary information. Bilateral and bimodal effects such as summation, head shadow and squelch, are general characteristics of bilateral hearing [11]. Moreover, bimodal hearing can offer the unique opportunity of combining complementary information by having access to two distinct modalities. Contralateral residual hearing is mainly situated in the low frequency region [12], which is known to contain cues regarding e.g. voice fundamental frequency, prosody and music [13–16]. Such information cannot well be captured by electrical stimulation. Literature shows that bimodal benefits can be demonstrated within the auditory domains of speech understanding in noisy situations, ease of listening, sound localization, music appreciation and sound quality [5, 17–27].

Despite the evidence of these bimodal advantages in research settings, the fact remains that multiple CI-recipients do not opt for the bimodal combination in daily life. Moreover, rather low correlations between objective performance measures and self-reported outcomes have been reported [28]. The authors didn't find this surprising since a laboratory environment provides only a selected sample of hearing abilities whereas self-ratings cover a generality of contexts in daily environments. Therefore, the self-assessed daily-life experiences of these patients are an important research area to address when investigating bimodal benefit.

Studies by Noble et al [28, 29] compared self-reported questionnaires amongst different profiles of CI-users (CI+CI, CI alone, CI+HA). Handicap ratings as well as hearing disability ratings in specific daily life listening situations, could not demonstrate significant differences between the bimodal group and the group with only a unilateral CI. This could suggest that the additional hearing aid does not contribute to improving the hearing ability or reducing the handicap perceived by unilateral CI-patients. In fact the bimodal group even scored slightly higher handicap ratings compared to the unilateral group. A more recent study [30] repeated the earlier study in a comparable group of bimodal and unilateral listeners using the same disability questionnaire. Even though scores seemed in favor of the bimodal users, again no statistical significant differences could be established between both groups, except for the scale of sound quality and naturalness. In addition, also a CI-related quality of life questionnaire was used in the same study. Results did show an improved rating of sound perception in bimodal compared to unilateral CI listeners.

Overall, questionnaire results so far seem to indicate that bimodal listeners experience only limited, if any, benefit over unilateral CI users. However in these studies only comparisons between groups were made.

### Current study

The current study aims to assess the experiences of recipients of a unilateral cochlear implant either wearing or not wearing a contralateral hearing aid by using a set of daily-life questionnaires in the field of bimodal use, hearing disability, hearing handicap and general quality of life. It was questioned whether comparisons between both groups could repeat the findings from previous literature [28–31]. Additionally, for the first time disability ratings across daily listening conditions were examined within the group of bimodal users. Up until now no study namely looked into comparisons between the condition with and without the hearing aid within the group of bimodal users as to address the perceived level of benefit.

## Materials and methods

### Ethics

The local Medical Ethical Committee (Maastricht University Medical Center, NL42011.068.13) has approved this study as part of a larger clinical trial registered in the Dutch National Trial Register (NTR3932). The study has been conducted in accordance with the ethical principles as formulated in the Declaration of Helsinki. All participants have provided written informed consent to the inclusion of their anonymous data and received a small participation gift (gadget package provided by Advanced Bionics™).

### Procedure

Inclusion criteria were that participants were capacitated adults, patients of CI-team South-East Netherlands, users of a unilateral CI of the brand Advanced Bionics™ (AB) (Valencia, USA), had at least one year of CI experience, used the CI speech processor more than 10 hours a day, were willing and able to fill out questionnaires and agreed to participate in the study by informed consent. Subjects were excluded if they were less than 18 years of age or incapacitated, were non-Dutch speaking or used bilateral cochlear implants.

All subjects that were deemed eligible, were invited by mail. They were requested to fill out the paper-based questionnaires and return them together with the informed consent form in order to participate. If no response, either positive or negative, was received within one month, a non-committal reminder was sent. When some of the responses of participating subjects were ascertained to be missing within or across questionnaires, a one-time request for clarification and addition was sent in order to complete the data collection.

### Questionnaires

A set of five self-administering questionnaires was compiled to assess daily-life experiences of unilateral CI recipients regarding their bimodal hearing aid use, hearing (dis)abilities, hearing handicap and health related quality of life (HRQL).

**Bimodal use.**   In order to explore the use of the bimodal hearing combination, a composite bimodal questionnaire was used. The bimodal questionnaire was designed in line with the questionnaire more recently used by Neuman et al. [32]. It represented a fusion of questions derived from four existing questionnaires: the bimodal questionnaires by Tyler et al. [33], Fitzpatrick et al. [34] and Fitzpatrick & Leblanc [31], and the International Outcome Inventory for Hearing Aids (IOI-HA) by Cox et al. [35]. The resulting explorative questionnaire and its composition can be found in S1 File. Questions were formulated in Dutch and divided in three main parts: the experience with HAs prior to receiving the CI (7 items), the decision-making process on retaining the contralateral HA (3 items) and the experiences with a contralateral HA after CI implantation (29 items). The main themes that were addressed by a combination

of open and closed questions are: the frequency of HA use, the situations in which bimodal stimulation is preferred, the fitting of CI and HA settings, the balance/fusion of CI and HA, the satisfaction with the HA and the decision of (dis)continued HA-use. The bimodal questionnaire was composed to be applicable to all unilateral CI-recipients whether they still wore a contralateral HA, tried one but stopped using it or did not try a contralateral HA at all. In the latter case a number of questions could be skipped in order to fit the patient-related situation.

**Hearing (dis)ability.**   Two relevant questionnaires were used to measure the patients' perception of residual hearing disability in daily-life listening situations: the SSQ and the AVETA. The SSQ, the Speech, Spatial and Qualities of hearing scale, was used in its Dutch version [36]. This questionnaire was designed and validated by Gatehouse & Noble [37] to reflect real life listening experiences of patients with different hearing profiles and rehabilitative interventions such as hearing aids and cochlear implants [28, 38]. The SSQ was designed with a special focus on daily listening situations whereby binaural auditory functions play an important role such as understanding speech in complex situations and localizing environmental sounds. The SSQ consists of 49 questions, divided in three main scales, asking subjects to respond on a visual analog scale of 0 (not at all) to 10 (perfectly). The Speech scale (14 items) asks to rate the ability to understand speech under quiet and noisy conditions as well as in more complex situations of speech perception. The Spatial scale (17 items) questions the performance to localize sounds and judge the distance of moving objects. Finally, the Qualities scale (18 items) aims to map the identification and segregation of sounds as well as the naturalness and effort of listening. In total, these substantives subdomains make that the questionnaire can also be described by 10 pragmatic subscales [39].

Secondly, the AVETA, Amsterdam Questionnaire for Unilateral or Bilateral Fittings [40], is a Dutch specialized questionnaire that combines two existing questionnaires: the Amsterdam Inventory of Auditory Disability and Handicap [41] and the Abbreviated Profile of Hearing Aid Benefit [42]. Originally the AVETA also included questions from the IOI-HA [35], however to avoid overlap with the questionnaire on bimodal use only the 18 main questions of the AVETA were used. The questionnaire covers six categories: detection of sounds (3 items), discrimination or recognition of sounds (3 items), speech intelligibility in quiet (3 items), speech intelligibility in noise (3 items), directional hearing and comfort of loud sounds (3 items). Subjects were asked to respond by checking one of four answering alternatives (never, sometimes, often, almost always).

For both hearing (dis)ability questionnaires subjects were asked to respond to each question as fitted with their daily hearing device or devices. Those subjects who continued to regularly use a contralateral HA besides the CI were asked not only to fill out each question for the bimodal condition but to also respond for the listening condition with CI alone and with HA alone in order to reflect the perceived added value of each device besides the other.

**Hearing handicap.**   According to the World Health Organization [43], handicap is defined as a disadvantage for a given individual, resulting from an impairment or a disability, that limits or prevents the fulfillment of a role that is normal for that individual. Handicap as such represents the continued socialization of an impairment or disability. To assess whether wearing a contralateral hearing aid affects the hearing specific perceived handicap of unilateral CI patients, the Hearing Handicap Questionnaire (HHQ) was used. The questionnaire was developed by Gatehouse and Noble [37] aside from the SSQ questionnaire and has been validated among CI recipients [29] as a useful questionnaire of the true concept of handicap as defined by the World Health Organisation [43]. The original English questionnaire was translated and back-translated into Dutch by a native English audiologist with a good level of Dutch and two native Dutch speakers with a Master's degree in English. The 12 questions of the HHQ can be resolved into two unique factors: emotional distress (7 items) and social

restriction (5 items). Subjects were asked to respond by checking one of five answering alternatives (never, rarely, sometimes, often, almost always) representing their daily perceived handicap as fitted with their hearing device(s).

**Health related quality of life.**   Hearing loss is known to affect the ability to exchange information and therefore affecting a person's quality of life [44]. In this study the Dutch version of the HUI3, Health Utility Index Mark III, was used to assess HRQL [45]. The questionnaire is a generic multi attribute preference-based measure of health status and HRQL that is widely used as an outcome measure in clinical studies [46]. Subjects were asked to check their daily perceived level of health with respect to 8 attributes: vision, hearing, speech, ambulation, dexterity, emotion, cognition and pain/discomfort. The levels within each attribute vary from highly impaired or disabled to normal. The attribute level per health domain was determined and consequently translated to the multi-attribute level utility score whereby each attribute is considered in relation to the other health domains. Finally, by applying health related weights and combining scores across all attributes, the resulting total score of HRQL on a scale of 0 (= death) to 1 (= full health) is calculated. The HUI3 has been put forward as being the instrument of first choice when measuring utility within a population with hearing complaints [45] and in CI-recipients [47].

## Sample size calculation

A pre hoc sample size estimation was performed based on the primary outcome of bimodal benefit: the benefit of wearing CI and HA together in comparison to only wearing a CI. Regarding hearing (dis)ability, measured by the SSQ, comparisons between different profiles of CI-users (bimodal versus unilateral) previously did not show any significant difference in literature [28]. The added aim of the current study however lies in investigating the difference between listening conditions within the same bimodal subject. Since regarding these comparisons no estimations are available from literature, a medium effect size of .50 was considered. This represents a difference of half a standard deviation as being clinically relevant and follows the recommendations made by Cohen [48]. For a two-sided paired sample t-test, with a significance and power level of respectively 0.05 and 80%, the sample size was estimated at 32 bimodal subjects. Taken into account a response rate of 75% and a bimodal retention rate of about 60% [9], the amount of CI recipients (not knowing in advance whether or not they currently are bimodal or unilateral users) to be approached was set at approximately 70.

## Participants

Between April 2013 and March 2014, invites were sent out to all 77 patients of the CI team South-East Netherlands who were judged to be eligible according to the inclusion criteria. As a result 48 subjects were included in the study by completing the questionnaires, while 29 subjects were not included due to reported or unreported non-response or incomplete informed consent. The response rate in this study (48/77, 62%), is somewhat lower than expected, but comparable to other questionnaire studies [30]. 26/48 subjects reported to regularly use (>50% of time) a conventional hearing aid in the contralateral ear (bimodal group), while the other 22/48 subjects only used the unilateral CI (unilateral group). All subjects, except one who used a Neptune processor, were users of an Harmony speech processor on the CI side (Advanced Bionics™). The bimodal subjects used a variety of conventional hearing aids in the non-implanted ear (14 Phonak™, 8 Oticon™, 2 Widex™, 1 Siemens™, 1 private label). Table 1 presents the mean characteristics and hearing history of patients in both groups alongside their last audiometric results within one year around the time of study involvement. The average duration of deafness and the amount of residual hearing pre-implantation is also displayed. Further details on patient characteristics can be consulted in S1 Table. Hearing history and

Table 1. Mean patient characteristics of bimodal and unilateral subjects.

| Variable | Bimodal | | Unilateral | | p |
|---|---|---|---|---|---|
| | (n = 26) | | (n = 22) | | |
| Sex (male/female,n) | 11/15 | | 15/7 | | 0.089 |
| Age (years) | 63.8 | (2.7) | 67.3 | (3.0) | 0.384 |
| Age onset deafness (years) | 21.7 | (4.3) | 30.8 | (5.0) | 0.173 |
| CI-experience (years) | 3.6 | (0.4) | 4.3 | (0.4) | 0.206 |
| Duration of deafness pre-implantation (years) | 38.8 | (3.2) | 32.1 | (3.5) | 0.164 |
| Residual hearing pre-implantation (PTA, dB HL) | 92.1 | (2.6) | 100.2 | (4.3) | 0.103 |
| Residual hearing (PTA, dB HL) | 96.6 | (2.7) | 107.6 | (3.8) | 0.020* |
| CNC FITTED (%) | 70.4 | (4.0) | 62.6 | (6.7) | 0.323 |
| CNC CI alone (%) | 58.7 | (4.4) | - | | - |
| CNC HA alone (%) | 46.3 | (4.4) | - | | - |

Mean (standard error) per group and significance of difference between groups (*p<0.05) based on Fisher's exact test for the variable 'sex' and independent-samples Student T-tests for other variables. CI = cochlear implant, HA = hearing aid, FITTED = referring to CIHA in case of bimodal and CI in case of unilateral, CNC = consonant-nucleus-consonant maximum phoneme score across 55-65-75 dB SPL in quiet free-field, PTA = pure-tone average across 0.5, 1 and 2kHz under headphones in the unaided non-implanted ear.

other demographic information appeared not to differ between both groups. The residual hearing in the non-implanted ear was better for the bimodal group compared to the unilateral group with an average difference of 8 dB pre-implantation and 11dB at the time of study involvement. The latter difference was statistically significant (p<0.05, independent-samples Student t-test) and appeared most pronounced for hearing thresholds up to 1000Hz (Fig 1). The difference at the time of CI-surgery failed to reach significance. Between implantation and time of study involvement (around 4 years) residual hearing thresholds deteriorated on average 7,3 dB (p = 0,020) in the unilateral group and 4,5 dB (p = 0,004) in the bimodal group (dependent-sample Student t-tests).

## Data analysis

Data were digitally entered by two independent persons using an Excel file designed to avoid invalid entries. The final data file arose by merging both entry versions and correcting for discrepancies. The data of the explorative questionnaire on bimodal use were handled descriptively. The data of the other questionnaires were analyzed statistically after dealing with missing values. Statistics were performed using IBM SPSS Statistics version 21.0.0.1.

Missing values were handled per questionnaire and per group. Cases were included if more than 90 percent of the questionnaire items was completed. Bimodal cases, when measured under three listening conditions (CI, HA, CIHA), were included if they had more than 90 percent complete data in at least one listening condition and up to a maximum of one listening condition without any item response. In the resulting dataset missing responses on questionnaire items were replaced using multiple imputation (MI) to generate 100 complete datasets. These datasets were then analyzed separately, and finally a single (pooled) MI estimate and its standard error was calculated by combining the estimates and standard errors obtained from each completed dataset using 'Rubin's rules' [49, 50]. The imputation method was 'Fully Conditional Specification' or 'Monotone' as determined by the automatic method in SPSS's module for multiple imputation. To improve the accuracy of the imputed values [51], next to questionnaire items the imputation model also included participants characteristics (Table 1) as auxiliary variables. Missing item responses were modeled by linear regression.

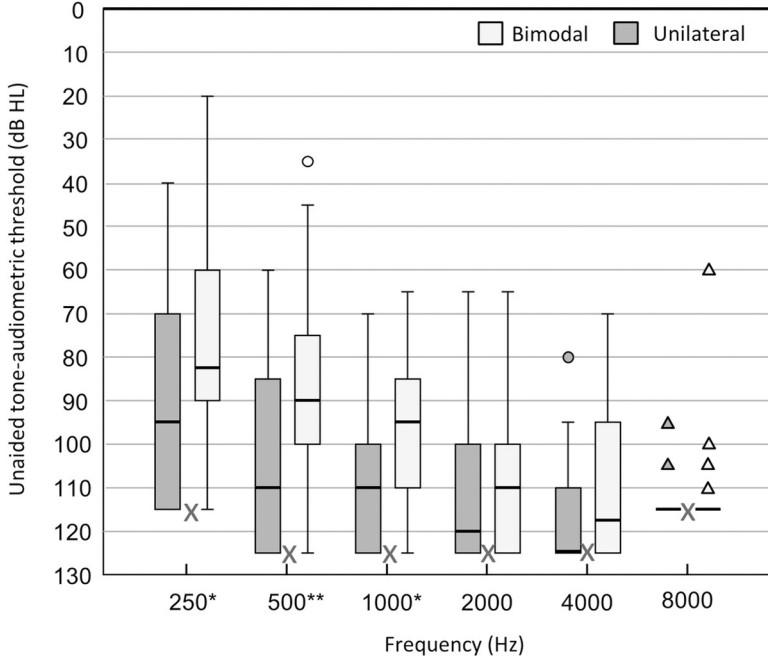

**Fig 1. Residual Hearing at study involvement.** Unaided pure-tone thresholds in the non-implanted ear for bimodal (n = 26) and unilateral (n = 22) subjects within one year around the time of study involvement. If no response could be recorded within the limits of the audiometer, a value of 5dB HL greater than the maximum tested level was entered (see X markings). Box plots represent the distribution per frequency (median and interquartile range), with whiskers denoting minimum and maximum values within 1.5 times the interquartile range, circles denoting outliers and triangles denoting extremes. Significant differences between groups are based on independent-samples Student T-tests per frequency (*p<0.05, **p<0.01).

To compare outcomes between groups parametric two-tailed independent samples student *t*-tests were conducted for the listening condition 'as fitted', referring to CI alone in case of the unilateral group and CI and HA together for the bimodal group. All subjects of the bimodal group rated their hearing disability using the SSQ and AVETA questionnaires under 3 conditions (CI, HA and CIHA). Since 3 observations (level 1) were clustered within subjects (level 2) the data had a two-level structure. To account for the hierarchical structure, marginal multilevel model analyses were applied. The restricted maximum likelihood ratio test was used to select the most appropriate covariance structure of the residuals (either an unstructured or a compound symmetry matrix). When a significant main effect was found, the three listening conditions (CI, HA and CIHA) were all compared pairwise and Bonferroni adjusted p-values were considered. Statistical significance is defined as a p-value of < 0.05.

## Results

### Bimodal use

The complete questionnaire on bimodal use can be consulted in S1 File. The absolute and relative frequency at which each answering alternative is chosen, are presented per question. Furthermore results are visually summarized as compared between the unilateral and the bimodal users. A descriptive overview of the main findings is presented here.

In both groups more than 60% of the participants used to regularly (>10hours a day) wear a conventional HA in both ears prior to receiving their cochlear implant (Q3). The degree of bilateral HA experience therefore does not seem to be related to whether subjects either

retained or rejected the contralateral hearing aid after implantation. The helping value of conventional hearing aids however does appear to be different between both groups (Q6). Most of the bimodal users (56%) experienced their preoperative HAs as being very helpful, while a large group of unilateral CI users (36%) described their HAs as rarely being helpful. This is in line with the finding that most of the bimodal users (65%) reported the prior intention of retaining the hearing aid aside the CI, while most unilateral users (68%) had not yet made up their mind before surgery on retaining a contralateral HA(Q8). As a result, 77% of subjects in the bimodal group, compared to only 27% in the unilateral group, started using the HA directly after CI surgery (Q15). In contrast, half of the subjects in the unilateral group never tried a contralateral HA after receiving their CI and the majority of unilateral subjects who did try a HA (78%) reported that the sound from both sides didn't fuse to become one image (Q34). Overall the use of a contralateral HA augmented (92%) the personal enjoyment in life for bimodal subjects while it made no difference (56%) or resulted in a degradation (33%) for subjects in the unilateral group (Q37).

## Hearing (dis)ability

**SSQ.** Two unilateral and two bimodal cases were excluded from the analysis of the SSQ questionnaire due to more than 10% missing values. The pooled overall score and the scores per main scale of the SSQ questionnaire are presented in Fig 2. Details on the mean scores of the ten pragmatic subscales of the SSQ questionnaire can be consulted in S1 Fig.

Results showed no significant difference of the overall score or any of the (sub)scales between the unilateral and the bimodal group (as fitted). Within the bimodal group however, different listening conditions (CI, HA, CIHA) did show significantly different scores (F(2,23); all p<0.001). Bimodal subjects rated the daily life performance of the combination of CI and HA together significantly higher (p<0.01) than the performance of the CI (about 1 point) or the HA (about 2 points) only. This finding was consistent across all daily auditory functioning situations as represented by scoring the different (sub)scales. The hearing ability of bimodal

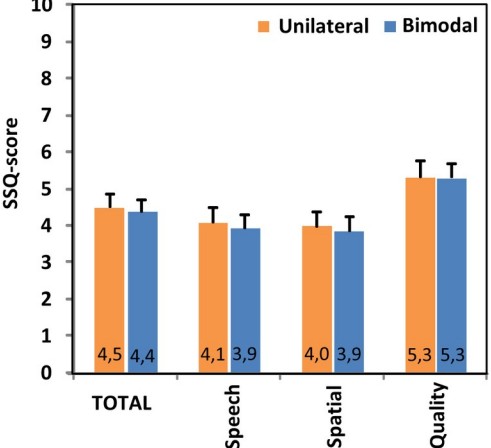
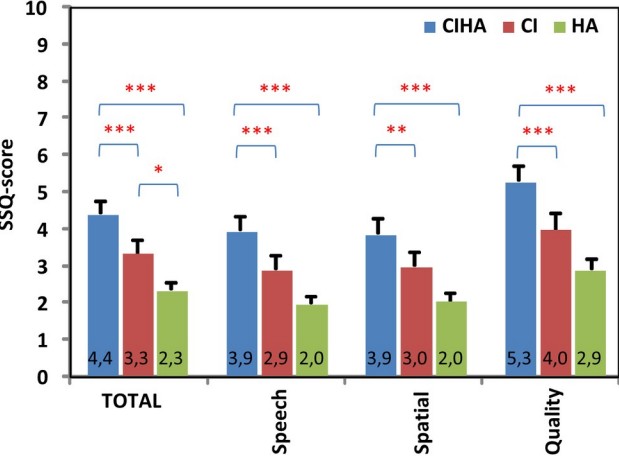

**Fig 2. SSQ.** Hearing (dis)ability ratings for subjects who only use a cochlear implant (CI) (unilateral group, n = 20) and subjects who also use a hearing aid (HA) in the non-implanted ear (bimodal group, n = 24). Pooled mean (+standard error) overall scores (SSQ) and scores per main scale (Speech, Spatial, Quality) on a visual-analogue scale (VAS, 0–10) using the SSQ-questionnaire by Gatehouse & Noble [37]. Scores were compared between groups as fitted (A.) and evaluated within the bimodal group for listening conditions with CI, with HA and with CI and HA together (B.) Asterisks denote significant differences between groups or listening conditions (*p<0.05, **p<0.01, ***p<0.001).

subjects in the CI alone condition was rated to be better compared to the HA alone condition across all subscales. This difference however only reached significance ($p<0,05$) on 3 subscales as well as on the overall SSQ outcome ($p<0,05$).

**AVETA.** Three unilateral and four bimodal subjects were not included in the final analysis of the AVETA questionnaire since more than 10% of their data were established to be missing. The pooled overall score and the scores per category on the AVETA questionnaire are presented in Fig 3. Although a slightly higher level of discomfort for loud sounds was seen in the bimodal group compared to the unilateral group, no significant difference was found

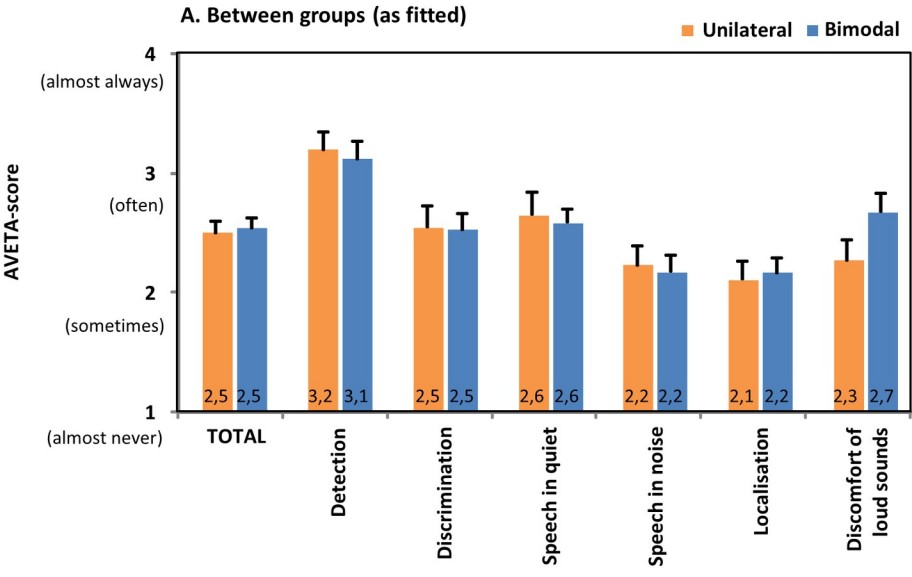

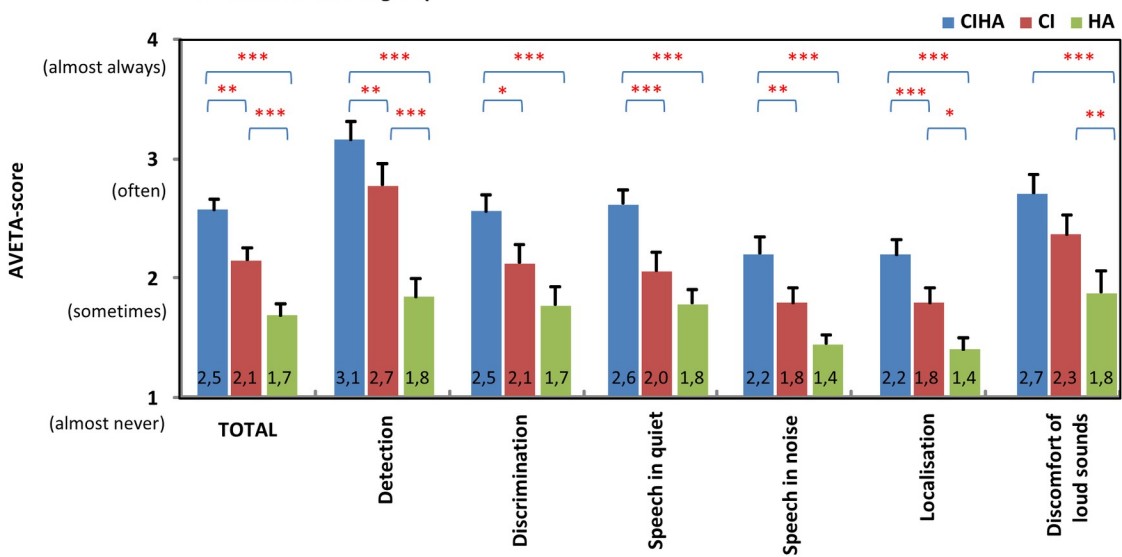

**Fig 3. AVETA.** Hearing (dis)ability ratings for subjects who only use a cochlear implant (CI) (unilateral group, n = 19) and subjects who also use a hearing aid (HA) in the non-implanted ear (bimodal group, n = 22). Pooled mean (+standard error) overall score and scores per category using the AVETA-questionnaire by Boymans et al. [40]. Scores were compared between groups as fitted (A.) and evaluated within the bimodal group for listening conditions with CI, with HA and with CI and HA together (B.) Asterisks denote significant differences between groups or listening conditions (*p<0.05, **p<0.01, ***p<0.001).

between both the unilateral and the bimodal group on the overall score or on any of the other subscales. For subjects within the bimodal group however, a significant effect of listening conditions (CI, HA, CIHA) was found ($F_{(2,42)}$; all $p < 0.001$). Listening with the bimodal combination (CIHA) was rated significantly higher compared to the CI or the HA alone ($p < 0.05$) in all questioned auditory situations except in the discomfort scale. The performance with CI alone was rated significantly better compared to HA alone for the overall scale as well as for the detection and the localization scale ($p < 0.05$). The discomfort level for loud sounds was rated significantly higher (it is less favorable) ($p < 0.05$) for the bimodal combination compared to the HA alone condition. The CI alone condition in its turn was experienced to be less comfortable ($p < 0.05$) than the HA alone condition.

## Hearing handicap

All cases were included in the analysis of the HHQ questionnaire and no imputations were performed since no missing values occurred. Results of the HHQ questionnaire are presented in Fig 4. Although ratings in the bimodal group were higher for the overall outcome as well as on both subscales, suggesting a slightly higher level of perceived handicap compared to the unilaterally fitted subjects, scores were not found to be significantly different between both groups.

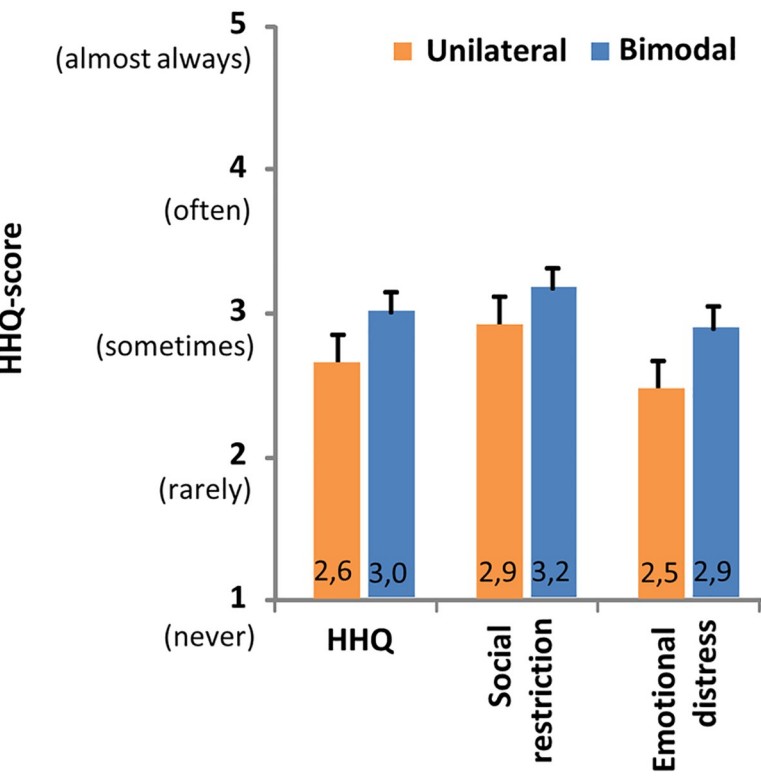

**Fig 4. HHQ.** Hearing handicap ratings for subjects who only use a cochlear implant (CI) (unilateral group, n = 22) and subjects who also use a hearing aid (HA) in the non-implanted ear (bimodal group, n = 26). The mean (+standard error) overall score and scores per subscale using the HHQ-questionnaire by Gatehouse and Noble [37]. No significant differences between groups (as fitted) were found (p>0.05).

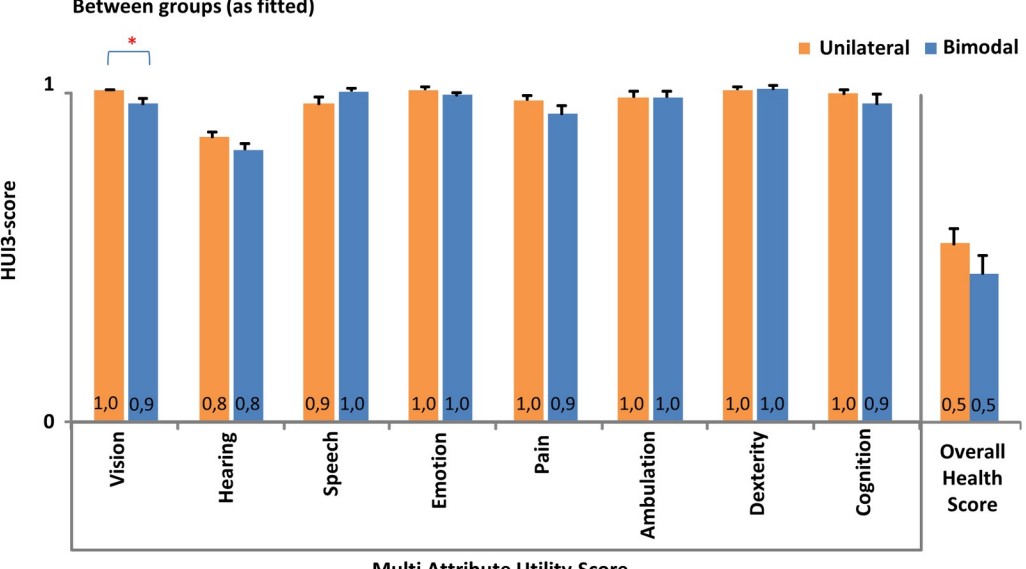

**Fig 5. HUI3.** Health related Quality of Life ratings for subjects who only use a cochlear implant (CI) (unilateral group, n = 21) and subjects who also use a hearing aid (HA) in the non-implanted ear (bimodal group, n = 26). Pooled mean (+standard error) multi attribute utility scores across health related subscales and the pooled mean (+standard error) overall health utility score are determined using the HUI3 questionnaire by Grutters et al. [45]. Asterisks denote significant differences between both groups (*p<0.05).

## Health related quality of life

One of the unilateral cases and none of the bimodal cases were discarded from the final analysis due to more than 10% missing values. The pooled multi-attribute utility score per attribute as well as the pooled overall HRQL score is presented in Fig 5. The attribute vision showed a small but significant difference between both groups (p<0.05), whereby the visual functionality in the bimodal group was scored less optimal compared to the unilateral group. No significant differences were found between both groups in any of the other health domains, including the hearing domain. It can be noticed that all health domains scored on average a perfect or near perfect attribute level, except for the hearing domain reflecting the population of hearing impaired subjects in the use of their hearing equipment. Although a slightly lower score was found in the bimodal group, no significant difference could be established between both groups looking at the overall HRQL.

## Discussion

### Experience with bimodal aiding

The aim of the current study was to assess the experiences of CI recipients either wearing or not wearing a contralateral hearing aid. The study sample consisted for 54% of bimodal subjects which seems a good representation of the target group, given that a bimodal retention rate of 64% was previously established within a larger sample (n = 77) of the CI population in the same CI center [9]. The experience of using the bimodal combination was examined by a qualitative questionnaire on bimodal use. Results of the questionnaire on bimodal use seemed to indicate that it is not the degree of hearing aid experience (i.e. 'hours of use') but rather the functional quality (i.e. 'helpfulness') of the hearing aid experience prior to implantation that differed between the bimodal and the unilateral group. A study by Fitzpatrick & Leblanc [31]

previously suggested a comparable conclusion. This finding can be interpreted in the light of the fact that demographic information and hearing history did not differ between participants in both groups. The degree of residual hearing in the contralateral ear was found to be better for the bimodal group compared to the unilateral group, especially at lower frequencies at the time of study involvement. A database study by Devocht et al. [9] demonstrated that the rate of retaining a hearing aid in the contralateral ear one year after receiving a unilateral CI, was not significantly related to demographics or hearing history. Instead, the retention rate was related to the amount of residual hearing, to the residual speech understanding score and to the difference in scores between the HA and the CI ear. The perceptual difference between both ears can in turn be linked to the current finding that a number of unilateral users reported an unfused sound image when trying the CI together with a HA.

Despite the given negative experiences of unilateral users who did try a contralateral HA, it should be noted that more than 50% never again tried a HA after receiving their CI. It is possible that a potential group of bimodal users was missed by never giving the HA a chance. Therefore, clinicians have an essential role in providing patients with the tailored fitting of the appropriate devices. When counseling eligible CI-candidates, it is clinically valuable to identify those subjects that would make good bimodal users. Previous research presented discrimination values to identify which unilateral CI patients are most likely to turn into bimodal users [9].

## Comparing bimodal and unilateral users

Daily life experiences of unilateral and bimodal CI users were measured by a set of self-administrative questionnaires, taking into account the different aspects of hearing impairment. To assess a subjects' personal health experience it is important to not only question a functional disability resulting from a physical impairment, but also ask about the social and cultural consequences of the impairment (handicap) as well as the overall health quality as perceived by the concerned individual [43, 52].

When comparing the outcomes 'as fitted' between those CI recipients who do (bimodal group) and those who do not (unilateral group) wear a contralateral HA, no significant differences could be found across scales concerning disability or handicap. This is in line with results of the previous studies by Noble et al. [28, 29] using the same questionnaires and finding no statistical differences between both groups either. Another study [30] did find higher scores for the bimodal compared to the unilateral group, however statistical significance could not be reached except for the SSQ-subscale of sound quality and naturalness. Current results showed a trend towards bimodal users rating this particular Quality subscale higher compared to the unilateral group. However, statistical significance could not be reached in the current sample. Other research has indeed demonstrated a statistically significant difference in sound percept by adding a HA aside a unilateral CI [26].

The overall health related quality of life was not found to be significantly different between both groups. The attribute level of vision by itself demonstrated a small statistically significant difference between both groups. Hereby the vision state of the bimodal users (0.98) was a little less optimal compared to the unilateral listeners (0,94). Such a small difference (0,04) would most likely not be considered clinically meaningful. If this finding however does represent a true underlying difference, one reason for this might be that a reduced visual quality urges to compensate by optimizing other sensory modalities, increasing the chance of wearing a contralateral HA. Low frequency acoustic hearing provided by a HA namely is known to improve the representation of voicing [13] which can compensate for reduced visual lip-reading abilities. Also, bimodal input can to some extent enable sound localization [17] which can make up

for a diminished visual orientation. However, no existing literature was found to support the current finding of a reduced vision state in bimodal users, indicating future research is warranted to investigate whether this finding is repeatable.

## Perceived bimodal benefit

The current study also questioned, to our knowledge for the first time, the differences in disability between the listening conditions with CI alone, HA alone and CI and HA together within the group of bimodal users. Results of the SSQ as well as the AVETA questionnaire showed that the bimodal combination of CI and HA together was rated significantly more favorable across all questioned daily life listening situations, except for discomfort of loud sounds (not significant). Concerning the aversiveness of loud sounds, a trend was observed towards a higher rating, that is less favorable, for the bimodal listening condition compared to the condition with HA alone or CI alone. This trend can be related to the programming of the devices and possibly can be explained in the light of the binaural loudness summation effect. Findings are consistent with the outcomes of a study by Boymans [53] using the same AVETA-questionnaire in users of bilateral conventional hearingaids. Hereby the bilateral combination resulted in more aversiveness of loud sounds compared to using only a unialteral HA. Giving these results it is important to include the restoration of normal loudness perception in the fitting process of hearing equipment. It has been demonstrated that loudness normalization especially of binaural broadband signals asks for individual gain corrections which cannot always be corrected for by compression algorithms [54].

The observation that statistically significant bimodal benefit was present in all daily life listening situations questioned by the SSQ and the AVETA, shows that bimodal hearing cannot only be beneficial in complex situations reflecting true binaural hearing capacities, but also means an improvement for basic auditory functions [30] like identifying sounds and understanding speech in quiet. It has been confirmed that bimodal aiding indeed enhances multiple dimensions of speech perception such as intelligibility, listening effort and sound quality, whereby the bimodal summation effect and the access to complementary information is believed to play an important role [26].

It is known that the degree of bimodal benefit in laboratory settings shows a large amount of variability among bimodal listeners [24, 26]. The chance of a unilateral CI-recipient becoming a bimodal user, is known to be related to the level of residual hearing [9, 30]. Meanwhile, the actual degree of bimodal benefit cannot be explained by the amount of residual hearing alone [26, 55–57]. For example, it is known that the fitting of the HA may play an important role [58, 59]. In the current study, the fitting of the CI and the HA were not assessed directly. Rather, the data show how subjects evaluated their daily life functioning by using their hearing devices in the daily settings. All subjects were active patients at the CI center of MUMC+ and had regular check-ups of their devices when necessary. In clinical practice, as in many CI centers around the world, no systematic bimodal fitting protocol was applied, since no generally accepted bimodal fitting method exists [7]. Furthermore other factors such as the spectral resolution ability of the contralateral ear [60] as well as the difference in speech recognition abilities between both ears [61] have been suggested as playing a role in the measured bimodal benefit. When looking at the degree of bimodal benefit in the current study an average amount of 0.6 to 1.9 points was seen across all SSQ-scales using a scaling from 0 to 10. For the AVETA-questionnaire a mean benefit of 0.3 to 0.6 was observed with a scaling from 1 to 4. The established degree of bimodal benefit lies in range with an average SSQ-score difference of 0.8 to 2.0 that is reported for subjects transferring from one CI to a successive second CI [62, 63]. The question arises as to what makes a bimodal and a unilateral CI-recipient different, and yet results in

a rather comparable disability outcome 'as fitted'? It has been hypothesized before that listeners choosing a bimodal fit might experience tougher demands in their daily life activities and may have not been performing well with the CI alone, giving rise to a continued HA use in an effort to improve things [28]. Bimodal subjects indeed rated their ability level with CI alone less optimal compared to the unilateral group. However, no significant difference in CNC-score with CI-alone was observed between both groups. Overall the bimodal group scored slightly better for the condition 'as fitted'. Earlier research indeed showed that the CNC-score with CI alone is in itself not related to bimodal HA retention [9]. There is no doubt however that daily auditory functioning is much more complex than illustrated by testing word scoring in quiet. It should therefore be noted that current results also demonstrated that, even though not significant, the bimodal subjects on average scored their residual hearing handicap about 0.4 points higher (that is worse) on a scale of 0 to 5 compared to the unilateral CI users. This is in line with previous findings done by Noble et al. [29]. In the light of these handicap ratings, it could be that the personal expectations and environmental requirements related to daily auditory functioning, are more demanding within the group of bimodal subjects compared to the unilateral group. Further research into this social field of bimodal hearing is warranted in order to test this hypothesis.

On the other hand a set of technical developments are on the go in order to improve the bimodal experience of bimodal subjects [57, 64, 65]. Advanced bimodal cooperation, entailing a combined fitting procedure of CI and HA, more enhanced inter-device communication and synchronized operation could enlarge the differences between bimodal and unilateral subjects even further. Measuring the amount of perceived bimodal benefit should be part of future investigations into the added value of these bimodal innovations.

## Methodological considerations

Since missing data are a familiar phenomenon in a survey study, efforts were undertaken to complete the data collection. For the SSQ and AVETA survey it was observed that missing items mainly occurred for the CI and HA alone conditions within the group of bimodal users. This might be due to the fact that most bimodal subjects wore the CIHA combination during most of the day, making it more difficult to judge unilateral listening conditions in itself. This is an important limitation of the current study set-up which by all known means could not be avoided. Bimodal subjects who at first did not respond to some unilateral conditions were therefore provided with the extra instructions to complete all conditions in order to reflect the added value of one device versus the other when listening across all questioned daily life situations.

Despite all efforts, however a portion of the data remained missing. Cases marked by more than 10% missing values were removed from the final analysis per questionnaire. As a result the sample size varied slightly across quantitative questionnaires. The amount of excluded cases fortunately remained limited and was comparable in the bimodal and unilateral group. Since no direct evidence was found to establish the origin of the missing values in the remaining cases, missing values were dealt with by means of multiple imputation taking demographic information into account. This method is especially designed for complex cases where theoretically satisfactory answers are difficult to derive explicitly and has proven its validity under the assumption of ignorable non response [49].

A survey study is known to be subjective in nature since it questions the personal experiences of subjects. By rating daily listening situations responses are made against an internal point of reference. When comparing hearing devices it is not possible to treat the conditions blindly. Especially in the case of bimodal subjects when questioning bimodal benefit by

comparing the listening conditions with CI, HA and CIHA, a bias towards the most used and preferred bimodal situation cannot be ruled out. It is therefore important to realize that the observed difference between the bimodal and the unilateral conditions should not be seen as a quantification of the actual measurable benefit but as the reflection of the perceived merit driving bimodal subjects to opt for the bimodal combination.

The small sample sizes of most CI studies (e.g. Devocht et al. [26]) often limit the ability to investigate correlations between the amount of bimodal benefit and related factors with sufficient statistical power. Combining findings by meta-analysis into larger samples and prospective follow-up of subjects whereby patients act as their own control, should therefore be considered in future research.

## Used questionnaires

Disability outcomes were collected using two questionnaires. The SSQ questionnaire is a well-established tool within the field of hearing rehabilitation and is available in multiple languages. With its 49 questions it is however a quite extensive survey to complete, making it not very suitable for clinical practice. The AVETA on the other hand only consists of 18 illustrated questions with confined answering alternatives. The survey has been developed with the same basic intentions as the SSQ, but is on the other hand only available in Dutch language. The current study illustrates that comparable results were achieved by using both questionnaires within the field of bimodal aiding. Since the start of the current study, a new version of the SSQ questionnaire, namely the SSQ12, has been proposed [66]. This reduced form of the SSQ-questionnaire is especially developed for clinical use. In this light current data were also evaluated on the proposed 12 items only (not displayed). Since the perceived bimodal benefit was present across all questioned listening situations, the SSQ12 outcome yielded comparable results compared to the full SSQ survey. Based on these findings it can be advised to use the SSQ12 survey when evaluating bimodal disabilities in a clinical setting.

The HUI3 is a general QoL questionnaire that is frequently used in and presented as the preferred QoL survey regarding hearing related research [45, 47, 67]. It should however be remarked that the questions reflecting the hearing item in itself are very general in nature and do not differentiate between hearing devices (e.g. hearing aid, cochlear implant, etc). This can lead to confusions especially in the case of bimodally fitted subjects. The cost-effectiveness of bimodal hearing devices versus bilateral implantation is gaining more and more interest these years [63, 68]. Therefore, it is important to use a QoL measure that is sufficiently sensitive to changes in hearing related health state, specifically concerning various listening conditions, as shown by the difference in daily life results within the current study's bimodal group.

## Conclusions

When investigating bimodal benefit, the self-assessed daily-life experiences of CI listeners are an important research area to address. The current study aimed at reviewing the experiences of unilateral CI recipients who do or do not retain a conventional hearing aid in the contralateral ear by a set of questionnaires on bimodal use, hearing disability, hearing handicap and general quality of life. At time of study involvement, subjects in the bimodal group had significantly more residual hearing below 1kHz in the non-implanted ear compared to the unilateral group. 77% of bimodal listeners started using a contralateral HA right away after receiving the CI, while 50% of unilateral listeners never tried a contralateral HA. It seemed that not the extent but the helping value of the experience with conventional hearing aids prior to receiving the CI differed between both groups. Daily hearing abilities, residual handicap and general quality of life were not significantly different between both groups, which is in line with

previous literature. However, when questioning bimodal subjects on the perceived merits of the bimodal combination compared to the CI or HA alone, bimodal benefits were consistently reported across all listening situations. This illustrates that bimodal hearing is not only perceived as valuable in complex situations, but instead can improve listening experiences based on all basic auditory functions. The finding that the overall outcome for hearing disability was comparable between those using or not using a contralateral HA, while the objective outcome with CI in itself was comparable in both groups and bimodal subjects reported a consistent benefit of the bimodal combination, points to the importance of comprehending what drives the subjective performance of a bimodal user. It seems that the expectations and demands on daily auditory abilities, personality aspects and the social framework surrounding these subjects asks for further research.

## Supporting information

**S1 Table. Details on patient characteristics.** Mean, standard error (SE), minimum (MIN) and maximum (MAX) per group of bimodal and unilateral subjects at time of study involvement (unless otherwise stated). CI = cochlear implant, HA = hearing aid, FITTED = referring to CIHA in case of bimodal and CI in case of unilateral, CNC = consonant-nucleus-consonant maximum phoneme score across 55-65-75 dB SPL in quiet free-field, PTA = pure-tone average across 0.5, 1 and 2kHz under headphones in the unaided non-implanted ear.
(XLSX)

**S1 File. Bimodal questionnaire.** Qualitative questionnaire on bimodal hearing aid use and bimodal experiences for subjects who discarded the hearing aid (HA) aside the cochlear implant (CI) (UNI = unilateral group, n = 22) and subjects who continued to use the HA in the non-implanted ear (BIM = bimodal group, n = 26). Instructions, questions and results are presented. The origin of each question is described by its reference in literature. Results are presented per group as absolute frequencies as well as percentages across answering alternatives. The overall number of responding subjects per question may differ depending on the completeness of data as well as the relevance of each question in a patient-related manner.
(PDF)

**S1 Fig. Pragmatic subscales SSQ.** Hearing (dis)ability ratings for subjects who only use a cochlear implant (CI) (unilateral group, n = 20) and subjects who also use a hearing aid (HA) in the non-implanted ear (bimodal group, n = 24). Pooled mean (+standard error) scores for the 10 pragmatic subscales on a visual-analogue scale (VAS, 0–10) using the SSQ-questionnaire by Gatehouse & Noble [37]. Scores were compared between groups as fitted (A.) and evaluated within the bimodal group for listening conditions with CI, with HA and with CI and HA together (B.) Asterisks denote significant differences between groups or listening conditions ($^*$p<0.05, $^{**}$p<0.01, $^{***}$p<0.001).
(TIF)

## Acknowledgments

We thank all respondents for the time and effort taken to fill out the questionnaires. We are grateful for the assistance of S. Otterbeck, M. Truyens and M. van Belle in the translation process of the handicap questionnaire, the help of T.L.C. Scheepers in conducting the data entry check, the contribution of F. Scherf (employed by Advanced Bionics) during the protocol development of this study. The contents of this paper were part of the PhD dissertation (July 2019) of the first author. Preliminary portions of these data were presented at the 9th Asia Pacific Symposium on Cochlear Implants and Related Sciences (Nov 2013, Hyderabad, India),

the 224th Annual Meeting of the Dutch Head and Neck Society (April 2014, Nieuwegein, The Netherlands), the 13th International Conference on Cochlear Implants and Other Implantable Auditory Technologies (June 2014, Munich, Germany), the 8th international Symposium on Objective Measures in Auditory Implants (October 2014, Toronto, Canada), the 18th Jaresta-gung of the Deutsche Gesellschaft für Audiologie (March 2015, Bochum, Germany) and the Phonak Conference 'Hearing loss and audiological technology between present and future' (June 2016, Milan, Italy). These (poster)presentations did not went through a peer-review process nor yielded any formal publication.

## Author Contributions

**Conceptualization:** Elke M. J. Devocht, Josef Chalupper, Robert J. Stokroos, Erwin L. J. George.

**Data curation:** Elke M. J. Devocht, A. Miranda L. Janssen.

**Formal analysis:** Elke M. J. Devocht, A. Miranda L. Janssen, Erwin L. J. George.

**Funding acquisition:** Josef Chalupper, Robert J. Stokroos, Erwin L. J. George.

**Investigation:** Elke M. J. Devocht.

**Methodology:** Elke M. J. Devocht, A. Miranda L. Janssen, Erwin L. J. George.

**Project administration:** Elke M. J. Devocht.

**Resources:** Josef Chalupper, Robert J. Stokroos, Erwin L. J. George.

**Supervision:** Robert J. Stokroos, Herman Kingma, Erwin L. J. George.

**Validation:** Elke M. J. Devocht.

**Visualization:** Elke M. J. Devocht.

**Writing – original draft:** Elke M. J. Devocht.

**Writing – review & editing:** A. Miranda L. Janssen, Josef Chalupper, Herman Kingma, Erwin L. J. George.

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
