## [Decision Letter · Decision Letter 0]

8 Jul 2020

PONE-D-20-12241

Self-Assessment of Unilateral and Bimodal Cochlear Implant Experiences in Daily Life

PLOS ONE

Dear Dr. Devocht,

Thank you for submitting your manuscript to PLOS ONE. After careful consideration, we feel that it has merit but does not fully meet PLOS ONE’s publication criteria as it currently stands. Therefore, we invite you to submit a revised version of the manuscript that addresses the points raised during the review process.

Please consider the comments of the third reviewer who mainly focused on  your statistical analysis. He has concerns on the validity of your statistical evaluation, so please do appropriate changes here, at least changes to the text passages and explain why you chose your evaluation approach.

We look forward to receiving your revised manuscript.

Kind regards,

Andreas Buechner, PhD

Academic Editor

PLOS ONE

Journal Requirements:

"The authors of this manuscript have read the journal's policy and have the following conflicts: the work of the first author (EMJD) in this investigator-initiated study was financially supported by a research grant from Advanced Bionics Inc. The second author (AMLJ) provided statistical support, made possible by a grant from the Dutch Heinsius-Houbolt foundation. The third author (JC) holds a scientific post in the Advanced Bionics European Research Center. For the remaining authors no conflicts were declared. The study was designed in close cooperation between MUMC+ and Advanced Bionics who also reviewed the final manuscript. Data collection, analysis and decision to publish were all solely accounted for by MUMC+."

We note that one or more of the authors are employed by a commercial company: Advanced Bionics Inc.

2.1. Please provide an amended Funding Statement declaring this commercial affiliation, as well as a statement regarding the Role of Funders in your study. If the funding organization did not play a role in the study design, data collection and analysis, decision to publish, or preparation of the manuscript and only provided financial support in the form of authors' salaries and/or research materials, please review your statements relating to the author contributions, and ensure you have specifically and accurately indicated the role(s) that these authors had in your study. You can update author roles in the Author Contributions section of the online submission form.

2.2. Please also provide an updated Competing Interests Statement declaring this commercial affiliation along with any other relevant declarations relating to employment, consultancy, patents, products in development, or marketed products, etc. 

3. We noted in your Acknowledgement Section that a portion of your manuscript may have been presented or published elsewhere.

 "Preliminary portions of these data were presented at the 9th Asia Pacific

 Symposium on Cochlear Implants and Related Sciences (Nov 2013, Hyderabad, India), the 224th

Annual Meeting of the Dutch Head and Neck Society (April 2014, Nieuwegein, The

Netherlands), the 13th International Conference on Cochlear Implants and Other Implantable

Auditory Technologies (June 2014, Munich, Germany), the 8th international Symposium on

Objective Measures in Auditory Implants (October 2014, Toronto, Canada), the 18th Jarestagung

of the Deutsche Gesellschaft für Audiologie (March 2015, Bochum, Germany) and the Phonak

Conference ‘Hearing loss and audiological technology between present and future’ (June 2016,

Milan, Italy). The contents of this paper were part of the PhD dissertation (July 2019) of the first

author."

Reviewers' comments:

Reviewer's Responses to Questions

**Comments to the Author**

1. Is the manuscript technically sound, and do the data support the conclusions?

Reviewer #1: Yes

Reviewer #2: Yes

Reviewer #3: Partly

2. Has the statistical analysis been performed appropriately and rigorously? 

Reviewer #1: Yes

Reviewer #2: Yes

Reviewer #3: No

3. Have the authors made all data underlying the findings in their manuscript fully available?

Reviewer #1: Yes

Reviewer #2: Yes

Reviewer #3: Yes

4. Is the manuscript presented in an intelligible fashion and written in standard English?

Reviewer #1: Yes

Reviewer #2: Yes

Reviewer #3: Yes

5. Review Comments to the Author

Reviewer #1: The group of bimodal patients surveyed represents one of the largest user groups among CI users and is therefore of fundamental interest. The compilation of the questionnaire seems well thought out and conclusively proven. The questions about spatial hearing are of particular interest. The selection of the static methods appears correct. The formation of subgroups in categories with different residual hearing might have provided further detailed insights into the generally inhomogeneous group of bimodal users. However, a larger number of respondents would also be desirable. Since your survey focuses on Advanced Bionics users, results with newer CI systems in combination with a Naida Link hearing aid would be of interest, since the signal preprocessing of the two systems is very similar and may therefore improve spatial hearing for bimodal users. An indication of this would be desirable as an outlook. The audiometer limit should be specified.

Reviewer #2: I. General

The present paper aims at investigating subjective experiences of unilateral cochlear implant (CI) users by means of questionnaires. Two groups have been explored: BIM, patients who wear a hearing aid (HA) on the contralateral side (n = 26), and UNI, patients who do not (n = 22).

In group BIM contralateral residual hearing is low (96.6 dB, table 1, @250 Hz 82 dB, figure 1). In UNI contralateral residual hearing is even worse (107.6 dB, table 1, @250 Hz 95 dB, figure 1). Between these groups speech comprehension (BIM: 70.4%, UNI: 62.6%) is not statistically significant different (table 1). Both groups show no difference in self-rated disability, hearing handicap or general quality of life. The authors want to find out if their results are in line with other studies [27-30] (line 88-89). They are!

As novelty of this study disability ratings within the group of bimodal users (condition with and without hearing aid) are pointed out (line 89-92). Ratings are better in the HA+CI condition compared to the CI or HA condition alone.

Sounding is good and the topic is discussed deeply, almost lengthily.

II. Specific

Line 60. Among these 11 citations you mentioned many studies with a small number of subjects. But a study with a large number of subjects is missing and should be included: Illg, A., Bojanowicz, M., Lesinski-Schiedat, A., Lenarz, T., & Büchner, A. (2014). Evaluation of the Bimodal Benefit in a Large Cohort of Cochlear Implant Subjects Using a Contralateral Hearing Aid. Otology & Neurotology, 35(9), e240–e244. In this study a statistically significant benefit was found for the bimodal condition in a BIM group of n = 141 subjects. It is interesting that residual hearing was much better comparing to your study group. When your BIM group is wearing CI only, speech comprehension decreases from 70.4% to 58.7%. If they are wearing HA only, still speech recognition scored 46.3%. How can such results be achieved despite of that low residual hearing? Is the group very heterogeneous? Please comment on that.

Line 64-67. These sentences are so similar to [27], p. 513, that it should be marked again as citation. Or e.g. replace “This is not surprising …” by “the authors didn’t find this surprising …”

Line 262-263: Since experience with CI was on average 4 years (table 1), question Q3 is related to a time where residual hearing might have been much better than values given in table 1. You wrote (line 206) that data in table 1 were retrieved „within one year around the time of study involvement“, i.e. at a time where they were already using the CI for about 4 years. You should discuss that and if this might have an impact on conclusions. It would be very interesting to show the amount of residual hearing in the non-implanted ear at the time of CI surgery.

Line 265. You are measuring “quality of hearing” by question Q6, whether the HA was helpful. I think there is a difference between quality and helpfulness and these terms should not be mixed up.

Line 276. According to Q37, you may also include the n=5 who perceived only a little improvement leading to a total of 92% of augmentation of enjoyment in life.

Line 291-293. “Although most (sub)scales did not demonstrate a difference between performance with CI and HA, …” From App2 and Fig 2B I understood that all (sub)scales within the bimodal group did (!) demonstrate a difference between performance with CI (CI was better) and HA, but this difference was significant in only 3 out of 10 (sub)scales (App2). Please comment on that.

Line 361. It would be very comfortable to read the size of your larger sample in parentheses.

Line 364. Please comment a bit more on the term “functional quality”. I think you are referring to question Q6, where you named it helpfulness.

Line 550-552. You wrote that it is “important to comprehend what drives a bimodal user to prefer the bimodal combination in comparison to a unilateral subject” Can you explain more clearly, why it is not simply the amount of residual hearing?

References [18] and [63] are identical; reference [53] is incomplete, but has nearly the same title like [18] and [63].

III. Some typos:

App1_Bimodal Questionaire. First line: ‘ead’ � aid

App1_Bimodal Questionaire. Question 30. ‘enfironment’ � environment

Line 255. „Statistical is defined as a p-value of < 0.05“ � What do you mean? statistical significance?

Line 460. Missing space; andyet � and yet

Reviewer #3: This is a study conducted from data that is "part of a clinical trial". Hence, although following strict CONSORT guidelines may not be needed, still some changes are expected. I list them below, and these needs to be addressed to make the paper statistically viable.

(a) Abstract: The abstract has statements like "...participants report a significant benefit of bimodal activity....", without any statement of the strength of the benefit (say, p-values, etc). This may leave the reader confused, as an example of providing half-baked (summary) information. Same goes when something is non-significant.

(b) Sample size/Power: I am also surprised to see that there is no mention on the sample size/power they are trying to play with, to get the desired effect size. What's the choice behing recruiting 77 subjects? The current study with the sample size employed may very well be regarded as a pilot study; please follow recommendations of proposing, or stating target sample sizes as in the reference below:

https://www.ncbi.nlm.nih.gov/pmc/articles/PMC3203750/

(c) Data analysis: Multiple imputation was conducted to fill missing values, which is OK. However, group comparisons directly used parametric two-tailed t-tests. What is the justification? Responses can/may not follow Gaussianity; was that checked? Then, why not available 2-sample nonparametric tests wwere used for a robust comparison.

Furthermore, I am entirely confused why a linear mixed model (LMM) was employed later? It is not clear whether we have subjects having measures for CI, HA and CIHA? LMMs are mostly used under longitudinal designs, with time as an important predictor, and where understanding changes of responses with time is the prime focus. I don't see any time component here (am I wrong?). The writeup needs to be clear on how repetitions are happening for each subject (authors are considering subject as a random effect). If that is the case, simply try a GEE (generalized estimating equation), which is often considered as robust to the choice of the initial covariance structure, although under an asymptotic situation. GEE can be implemented in any standard statistical software. A likelihood-based LMM appeared unnecessary to me here.

The data analysis section should clearly state the reasons why a GEE was done as well, explaining the nature of the clustering in the data, etc.

6. PLOS authors have the option to publish the peer review history of their article (what does this mean?). If published, this will include your full peer review and any attached files.

Reviewer #1: No

Reviewer #2: No

Reviewer #3: No

---

## [Author Response · Author response to Decision Letter 0]

7 Oct 2020

The attached rebuttal letter contains a point by point answer to the rieviewer comments. Find here below a copy of these contents: 

RESPONSE TO REVIEWERS

PONE-D-20-12241

Self-Assessment of Unilateral and Bimodal Cochlear Implant Experiences in Daily Life

REVIEWER#1

The group of bimodal patients surveyed represents one of the largest user groups

among CI users and is therefore of fundamental interest. The compilation of the questionnaire

seems well thought out and conclusively proven. The questions about spatial hearing are of

particular interest. The selection of the static methods appears correct. The formation of subgroups in categories with different residual hearing might have provided further detailed insights into the generally inhomogeneous group of bimodal users. However, a larger number of respondents would also be desirable. Since your survey focuses on Advanced Bionics users, results with newer CI systems in combination with a Naida Link hearing aid would be of interest, since the signal preprocessing of the two systems is very similar and may therefore improve spatial hearing for bimodal users. An indication of this would be desirable as an outlook. The audiometer limit should be specified.

- The authors agree that looking into the amount of residual hearing as a factor related to the questionnaire outcomes of bimodal benefit would be very interesting. The number of subjects however limits the possibilities of performing correlations or creating subgroups with enough statistical power. Various studies however have demonstrated that residual hearing in itself only accounts for a small part of the variance that is observed in bimodal benefit. This was elaborated upon in the manuscript as part of the discussion sections ‘perceived bimodal benefit’ and ‘methodological considerations’. 

- The study population indeed is restricted to users of an Advanced Bionics cochlear implant system. This waives the type of CI system as a possible confounding factor. Since the conduct of this study (2013-2014) newer bimodal systems indeed were released on the market focusing on combining the fitting and information across CI and HA for bimodal users. As a general outlook on further developments of these systems an extra paragraph was added at the end of the discussion section ‘perceived bimodal benefit’ in the new version of the manuscript. 

- The limit of the audiometer +5dB is displayed by “x” in Figure 1.

 

REVIEWER#2

I. General

The present paper aims at investigating subjective experiences of unilateral cochlear implant (CI) users by means of questionnaires. Two groups have been explored: BIM, patients who wear a hearing aid (HA) on the contralateral side (n = 26), and UNI, patients who do not (n = 22).

In group BIM contralateral residual hearing is low (96.6 dB, table 1, @250 Hz 82 dB, figure 1). In UNI contralateral residual hearing is even worse (107.6 dB, table 1, @250 Hz 95 dB, figure 1). Between these groups speech comprehension (BIM: 70.4%, UNI: 62.6%) is not statistically significant different (table 1). Both groups show no difference in self-rated disability, hearing handicap or general quality of life. The authors want to find out if their results are in line with other studies [27-30] (line 88-89). They are!

As novelty of this study disability ratings within the group of bimodal users (condition with and

without hearing aid) are pointed out (line 89-92). Ratings are better in the HA+CI condition

compared to the CI or HA condition alone.

Sounding is good and the topic is discussed deeply, almost lengthily.

- The authors want to thank the reviewer for his summary and endorsement of our findings. 

II. Specific

Line 60. Among these 11 citations you mentioned many studies with a small number of subjects. But a study with a large number of subjects is missing and should be included: Illg, A., Bojanowicz, M., Lesinski-Schiedat, A., Lenarz, T., & Büchner, A. (2014). Evaluation of the Bimodal Benefit in a Large Cohort of Cochlear Implant Subjects Using a Contralateral Hearing Aid. Otology & Neurotology, 35(9), e240–e244. In this study a statistically significant benefit was found for the bimodal condition in a BIM group of n = 141 subjects. It is interesting that residual hearing was much better comparing to your study group. When your BIM group is wearing CI only, speech comprehension decreases from 70.4% to 58.7%. If they are wearing HA only, still speech recognition scored 46.3%. How can such results be achieved despite of that low residual hearing? Is the group very heterogeneous? Please comment on that. 

- Thank you for the suggested reference. The reference of Illg et al (2014) was added to the corresponding lines in the revised version of the manuscript. 

- The residual hearing and CNC scores indeed showed a broad range of outcomes within each group. This heterogeneity of both groups was a given that is often seen among CI recipients. To provide the reader of our manuscript with all the information on our patient population, a supplemental table (S1 Table) with details on patient characteristics was added to the new version of the manuscript.

Line 64-67. These sentences are so similar to [27], p. 513, that it should be marked again as citation. Or e.g. replace “This is not surprising …” by “the authors didn’t find this surprising …” 

- We agree and changed the wording of this sentence to “The authors didn’t find this surprising…” in the revised version of the manuscript.

Line 262-263: Since experience with CI was on average 4 years (table 1), question Q3 is related to a time where residual hearing might have been much better than values given in table 1. You wrote (line 206) that data in table 1 were retrieved „within one year around the time of study

involvement“, i.e. at a time where they were already using the CI for about 4 years. You should

discuss that and if this might have an impact on conclusions. It would be very interesting to show the amount of residual hearing in the non-implanted ear at the time of CI surgery.

- The authors appreciate the suggestion of the reviewer to incorporate the amount of residual hearing prior to CI –surgery. Indeed the first part of the bimodal questionnaire (Q1-7) reflects the situation prior to receiving a CI, while the first version of the manuscript only presented the residual hearing at the time of study involvement (around 4 years later in time). The residual hearing pre-implantation was therefore added to the new version of the manuscript as part of Table 1 and S1 Table. Prior to surgery, the difference in residual hearing between both groups was about 8dB in favor of the bimodal group. This difference failed to reach significance at that point. However it increased to 11dB and therefore showed to be significant at the time of study involvement. The decline in hearing between CI surgery and the time of study involvement was hence also presented for both groups. The sections of the manuscript where these findings were presented and discussed were adjusted accordingly.

Line 265. You are measuring “quality of hearing” by question Q6, whether the HA was helpful. I think there is a difference between quality and helpfulness and these terms should not be mixed up.

- The authors agree with the raised confusion between both wordings. We replaced “quality of hearing” by “helping value” to keep as close as possible to the wording of the related question (Q6). This was done in the results, discussion and conclusion sections of the new version of the manuscript. 

Line 276. According to Q37, you may also include the n=5 who perceived only a little improvement leading to a total of 92% of augmentation of enjoyment in life.

- Thank you for the suggestion. Indeed we took a conservative approach by only counting for the larger improvements, but also a little improvement (n=5) is an augmentation. Therefore we corrected 73% to 92% in the new version of the manuscript. 

Line 291-293. “Although most (sub)scales did not demonstrate a difference between performance with CI and HA, …” From App2 and Fig 2B I understood that all (sub)scales within the bimodal group did (!) demonstrate a difference between performance with CI (CI was better) and HA, but this difference was significant in only 3 out of 10 (sub)scales (App2). Please comment on that.

- The authors agree that the wording of this section of the results could raise confusion. Indeed the performance with CI was rated better compared to the HA across all (subscales). This difference however did not reach significance expect for 3 subscales as well as the overall score. This section of the results was provided with new wording in the new version of the manuscript. 

Line 361. It would be very comfortable to read the size of your larger sample in parentheses.

- The size of the study population in the reference paper was added between brackets (n=77) in the new version of the manuscript.

Line 364. Please comment a bit more on the term “functional quality”. I think you are referring to question Q6, where you named it helpfulness.

- Indeed the degree of hearing aid experience refers to ‘hours of use’ in Question 3 and functional quality refers to ‘helpfulness’ of the HA in Question 6. These explanatory terms were added between brackets in order to improve clarity within the new version of the manuscript. 

Line 550-552. You wrote that it is “important to comprehend what drives a bimodal user to prefer the bimodal combination in comparison to a unilateral subject” Can you explain more clearly, why it is not simply the amount of residual hearing? 

- Residual hearing is known to be an important factor that drives CI recipients to (not) retain a contralateral HA (Devocht, 2015). If a CI recipient becomes a bimodal user, the current results suggest that the subjective performance of these subjects is rated within a different perspective compared to a unilateral user. 

The authors acknowledge that the wording of the conclusion section raised unintended confusion towards the factors driving bimodal use a priori (i.e. residual hearing). Instead these lines of the manuscript are intended to point towards the importance of factors influencing the post hoc subjective performance of bimodal users (i.e. objective performance, subjective benefit and expectations/demands/…). The wording therefore was revised in the new version of the manuscript. 

Devocht, E. M. J., George, E. L. J., Janssen, A. M. L., & Stokroos, R. J. (2015). Bimodal Hearing Aid Retention after Unilateral Cochlear Implantation. Audiology and Neurotology, 20(6), 383–393. https://doi.org/10.1159/000439344

References [18] and [63] are identical; reference [53] is incomplete, but has nearly the same title like [18] and [63].

- Thank you for this detailed check of the reference list! References 18, 53 and 63 were by mistake indeed the same. They are now fused together in the new version of the manuscript.

III. Some typos:

App1_Bimodal Questionnaire. First line: ‘ead’ � aid

App1_Bimodal Questionnaire. Question 30. ‘enfironment’ � environment

- Thank you for indicating these typos. They were corrected in the new version of S1. 

Line 255. „Statistical is defined as a p-value of < 0.05“ � What do you mean? statistical significance?

- Indeed the word “significance” was missing here. It was added in the new version of the manuscript.

Line 460. Missing space; andyet � and yet

- Indeed a spacing was missing here. It was added in the new version of the manuscript.

REVIEWER#3

This is a study conducted from data that is "part of a clinical trial". Hence, although

following strict CONSORT guidelines may not be needed, still some changes are expected. I list them below, and these needs to be addressed to make the paper statistically viable.

(a) Abstract: The abstract has statements like "...participants report a significant benefit of bimodal activity....", without any statement of the strength of the benefit (say, p-values, etc). This may leave the reader confused, as an example of providing half-baked (summary) information. Same goes when something is non-significant.

- It is not encompassing to mention p-values in abstracts since these values can only be interpreted when more detailed information on the examined comparisons and the used data analysis is known. These details however lead to far from the dense contents of an abstract. However, we agree that mentioning “significance” without further information can also be misleading. Therefore we removed this wording form the abstract in the new version of the manuscript. 

(b) Sample size/Power: I am also surprised to see that there is no mention on the sample size/power they are trying to play with, to get the desired effect size. What's the choice behind recruiting 77 subjects? The current study with the sample size employed may very well be regarded as a pilot study; please follow recommendations of proposing, or stating target sample sizes as in the reference below: https://www.ncbi.nlm.nih.gov/pmc/articles/PMC3203750/

- Thank you for this critical note on the size of our patient sample. A pre hoc sample size calculation was performed as part of the study protocol when filing for ethical approval. As a result the study aimed at including approximately 70 CI recipients. This sample size calculation was not mentioned in the previous version of the manuscript, but now is added to the new version of the manuscript as a separate paragraph within the section Materials and Methods. 

(c) Data analysis: Multiple imputation was conducted to fill missing values, which is OK. However, group comparisons directly used parametric two-tailed t-tests. What is the justification? Responses can/may not follow Gaussianity; was that checked? Then, why not available 2-sample nonparametric tests were used for a robust comparison. 

- Indeed two-tailed t-tests were used to compare between groups. In this case the outcome measure is quantitative and no questionable results (e.g. outliers) were seen. We based our choice to use parametric tests on the "central limit theorem” given the fact that the sample size of both groups was >20 (namely n=26 for the bimodal group and n=22 for the unilateral group). The relevant guidelines by Moore et al (2007) state this theorem as follows: 

T procedures are quite robust against non-normality of the population. T-tests can therefore be used even for clearly skewed distributions when the sample is large (n ≥ 40 for one-sample t-tests). Since two-sample t procedures are even more robust than the one-sample t methods, the guidelines can be adapted from one to two-sample procedures by replacing “sample size” with the “sum of the sample sizes” (n1 + n2 >40). 

Moore, D. S., McCabe, G. P., & Craig, B. (2007). Introduction to the practice of statistics. (6th Ed.). W.H. Freeman & Co Ltd.

Furthermore, I am entirely confused why a linear mixed model (LMM) was employed later? It is not clear whether we have subjects having measures for CI, HA and CIHA? LMMs are mostly used under longitudinal designs, with time as an important predictor, and where understanding changes of responses with time is the prime focus. I don't see any time component here (am I wrong?). The write-up needs to be clear on how repetitions are happening for each subject (authors are considering subject as a random effect). If that is the case, simply try a GEE (generalized estimating equation), which is often considered as robust to the choice of the initial covariance structure, although under an asymptotic situation. GEE can be implemented in any standard statistical software. A likelihood-based LMM appeared unnecessary to me here. 

The data analysis section should clearly state the reasons why a GEE was done as well, explaining the nature of the clustering in the data, etc.

- Thank you for pointing out towards this point of the statistical analysis description. We have indeed misstated our analysis as a linear mixed model analysis in the manuscript. This was a confusion in wording, since in fact we performed a marginal multilevel model analysis. The description of the performed data analysis has therefore been changed in the new version of the manuscript (see last paragraph of the data analysis section). A more specific motivation of the performed data analysis is stated here below.

Since all subjects of the bimodal group rated their hearing disability using the SSQ and AVETA questionnaires under 3 conditions (CI, HA and CIHA) at almost the same moment, the design of the study was not longitudinal. However, since 3 observations (level 1) were clustered within subjects (level 2) the data had a two-level structure. To account for this hierarchical structure of the data, we applied a marginal multilevel model analysis using the Linear Mixed Models procedure (MIXED) option in SPSS. In the MIXED procedure the covariance structure of the residuals was specified and the likelihood ratio test (LR-test) was used to select the most appropriate structure (unstructured or compound symmetry). As correctly stated by reviewer 3, marginal models, also called population averaged models, are often estimated by solving generalized estimating equations (GEE). While there are many advantages to using GEE, including robustness to misspecification of the initial covariance structure, one considerable disadvantage is that estimating functions are not likelihood based and as a consequence, likelihood-based methods are not possible. In contrast, likelihood-based marginal multilevel models embrace the interpretation and robustness of regression coefficients from a marginal model, while retaining the likelihood inference capabilities such as the likelihood ratio test to select the most appropriate covariance structure (Griswold et al. ,2013). Therefore, we preferred estimating marginal multilevel models which, in case of an unstructured or compound symmetry covariance matrix, are in fact comparable to GEE marginal models with an unstructured respectively exchangeable structure. 

Griswold, M. E., Swihart, B. J., Caffo, B. S., & Zeger, S. L. (2013). Practical Marginalized Multilevel Models. Stat, 2(1), 10.1002/sta4.22. https://doi.org/10.1002/sta4.22

ADDITIONAL REQUIREMENTS

- Thank you for providing the reference materials for formatting the manuscript to PLOS ONE’s standards. All formatting changes were applied in the new version of the manuscript, the separate title page and the naming of all figures, tables and supplemental information.

2. Thank you for stating the information in the Competing Interests section.

We note that one or more of the authors are employed by a commercial company: Advanced Bionics Inc.

2.1. Please provide an amended Funding Statement declaring this commercial affiliation, as well as a statement regarding the Role of Funders in your study

2.2. Please also provide an updated Competing Interests Statement declaring this commercial affiliation along with any other relevant declarations relating to employment, consultancy, patents, products in development, or marketed products, etc. Please include both an updated Funding Statement and Competing Interests Statement in your cover letter. We will change the online submission form on your behalf.

-According to the e-mail correspondence with Plos One’s Editor (Kate Armal), the funding statement and competing interests statement were added to the revised version of the cover letter.

3. Please clarify whether this conference proceeding or publication was peer-reviewed and formally published.

- The mentioned (poster)presentations at conferences did not went through a peer review process nor yielded any formal publication. This statement was added to the acknowledgments section of the revised version of the manuscript as well as the revised version of the cover letter.

---

## [Decision Letter · Decision Letter 1]

11 Nov 2020

Self-Assessment of Unilateral and Bimodal Cochlear Implant Experiences in Daily Life

PONE-D-20-12241R1

Dear Dr. Devocht,

We’re pleased to inform you that your manuscript has been judged scientifically suitable for publication and will be formally accepted for publication once it meets all outstanding technical requirements.

Kind regards,

Andreas Buechner, PhD

Academic Editor

PLOS ONE

Reviewers' comments:

Reviewer's Responses to Questions

**Comments to the Author**

1. If the authors have adequately addressed your comments raised in a previous round of review and you feel that this manuscript is now acceptable for publication, you may indicate that here to bypass the “Comments to the Author” section, enter your conflict of interest statement in the “Confidential to Editor” section, and submit your "Accept" recommendation.

Reviewer #1: All comments have been addressed

Reviewer #2: All comments have been addressed

Reviewer #3: All comments have been addressed

2. Is the manuscript technically sound, and do the data support the conclusions?

Reviewer #1: Yes

Reviewer #2: Yes

Reviewer #3: (No Response)

3. Has the statistical analysis been performed appropriately and rigorously? 

Reviewer #1: Yes

Reviewer #2: Yes

Reviewer #3: (No Response)

4. Have the authors made all data underlying the findings in their manuscript fully available?

Reviewer #1: Yes

Reviewer #2: Yes

Reviewer #3: (No Response)

5. Is the manuscript presented in an intelligible fashion and written in standard English?

Reviewer #1: Yes

Reviewer #2: Yes

Reviewer #3: (No Response)

6. Review Comments to the Author

Reviewer #1: From my point of view, the author has taken up all the points in question and has sufficiently addressed the proposed changes. The note regarding the audiometer limit meant the maximum sound level that the audiometer can produce. The possible increment was not meant. Basically, that was just a side note that didn't necessarily have to be included.

Reviewer #2: (No Response)

Reviewer #3: (No Response)

7. PLOS authors have the option to publish the peer review history of their article (what does this mean?). If published, this will include your full peer review and any attached files.

Reviewer #1: No

Reviewer #2: No

Reviewer #3: No

---

## [Editor Report · Acceptance letter]

23 Nov 2020

PONE-D-20-12241R1 

Self-assessment of unilateral and bimodal cochlear implant experiences in daily life 

Dear Dr. Devocht:

I'm pleased to inform you that your manuscript has been deemed suitable for publication in PLOS ONE. Congratulations! Your manuscript is now with our production department. 

Kind regards, 

on behalf of

Professor Andreas Buechner 

Academic Editor

PLOS ONE